# The Influential Factors of the Habitat Quality of the Red-Crowned Crane: A Case Study of Yancheng, Jiangsu Province, China

**Yuxun Wang, Liang Fang \*, Chao Liu, Lanxin Wang and Huimei Xu**

School of Economics and Management, Beijing Forestry University, Beijing 100083, China; ellipsism1212@bjfu.edu.cn (Y.W.); liuchao2345@bjfu.edu.cn (C.L.); wlx187@bjfu.edu.cn (L.W.); xhm7220669@bjfu.edu.cn (H.X.)
* Correspondence: fl2021@bjfu.edu.cn; Tel.: +86-010-62338410

**Abstract:** In order to effectively protect the habitat of cranes, this study constructs an indicator evaluation system based on the ecology–economy–society complex system and adopts the comprehensive "entropy weight method and analytic hierarchy process" evaluation model and coupled coordination model to scientifically measure the degree of coordinated development of the EES system in Yancheng. Further, a negative binomial regression model based on LASSO was used to analyze the key factors affecting the habitat quality of red-crowned cranes, and a support vector regression model was used to predict the population size of the cranes. The results show that the degree of the coordinated development of the EES system exhibited a fluctuating upward phenomenon, and the population size of the cranes also had a similar evolutionary trend, which indicates that the interaction between the two was significant and that the degree of the coordinated development of the system had a positive impact on the quality of the habitat of the cranes. Three types of ecological indicators (normalized difference vegetation index, annual precipitation, and soil erosion area) and three types of social indicators (natural growth rate, rural Engel coefficient, and public library collection) are the key factors affecting the population size of the cranes. The prediction results of the support vector regression model showed that the population of the cranes showed a fluctuating upward trend during the prediction interval, with a maximum of 952 cranes and an overall growth rate of 69.70%. The population size of the cranes is related to human social activities and the surrounding ecological environment, and the main reason for the decline in the population size of the cranes is the destruction of the local vegetation cover by the rapidly growing population and frequent human activities. Therefore, to improve the habitat quality of the cranes, local government departments need to strengthen the publicity of wildlife conservation, reduce agricultural land reclamation and pesticide pollution, and promote the coordinated development of the EES system in the Yancheng area.

**Keywords:** red-crowned crane; EES; entropy weight method; analytic hierarchy process; coupling coordination degree; RS-SVR

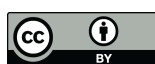

## 1. Introduction

Red-crowned cranes are the second-rarest large migratory birds in the world and have been listed as one of the world's priority species for rare wildlife protection [1]. The habitat quality of cranes is influenced by numerous factors, such as climate, water environment, and land use [2,3]. The disturbance to and destruction of their habitat caused by human activities are significantly increasing, which has led to the species recently becoming critically endangered [4]. Nevertheless, the available literature on the factors influencing the habitat quality of the red-crowned crane is still incomplete, and their comprehensive quantitative models still need to be systematically constructed in detail and practically validated, so further in-depth studies are required.

At present, habitat studies on these cranes in Yancheng, Jiangsu Province, have mainly focused on the Yancheng Nature Reserve, and various methods and approaches were used to investigate and analyze the habitat of these cranes. Research on the habitat distribution of the cranes in Yancheng Nature Reserve has shown that the cranes are relatively concentrated in the south beach and top hill, as well as the reeds, and have a certain habitat selectivity [5]. In addition, a study of the current distribution status and its trends in Yancheng demonstrated that the coastal mudflats of Yancheng are an important wintering habitat of the cranes, and the distribution is mainly concentrated in Dongtai, Dafeng, and Jianghu [6]. In addition, through continuous monitoring of the characteristics of the agricultural activities and the population of the cranes in Yancheng Nature Reserve, it was found that agricultural activities had some influence on the roosting and overwintering of the cranes. For example, the cranes preferred relatively dry cultivated fields, such as rice paddies and bean and wheat fields, when choosing their roosting sites, while preferring wetland habitats such as meadows in cases of flooding [7]. Remote sensing technology and GIS spatial analysis methods were used to study the influencing factors of the habitat of cranes, and the impact of changes in the wetland landscape on the habitat of the cranes was significant, mainly in terms of habitat area reduction, habitat quality decline, and food resource reduction [8]. In addition, research on the habitat selection and influence mechanisms of the cranes based on GPS positioning and ecological survey methods highlighted that the Yancheng Binhai wetlands are one of the main wintering sites of the cranes, and the distribution of these cranes in the wetlands showed a clear trend of concentration. The selection of habitat by the cranes is influenced by various factors, among which salinity variation is the main ecological factor affecting the selection of habitat by the cranes [9].

In addition to studies on habitat distribution and influencing factors, research on the influence of external systems on the habitat of the cranes also exists. For example, a study on the relationship between conservation of the habitat of the cranes and economic development shows that during the rapid economic development of Yancheng, the ecological environment of the wetlands was seriously damaged and threatened, and the survival conditions of wildlife, including the cranes, were affected [10]. Another study on changes in the patterns of the environmental system of the Yancheng coastal wetlands and their effect on the habitat of the cranes shows that these environmental changes had obvious effects on the habitat of waterbirds, including cranes, leading to habitat fragmentation, reduced food supply, and decreased habitat quality [11].

The abovementioned studies have conducted comprehensive investigations and analyses on the habitat of the red-crowned crane; however, there are still shortcomings. For example, in terms of habitat index selection, most of the previous research evaluated habitat quality based on habitat factors [12]. It is worth noting, however, that the habitat conservation of the red-crowned crane is not an independent issue but rather an interdependent one with the ecological, economic, and social systems of surrounding habitat areas [13,14]. Moreover, in terms of indicator assignment methods, due to the complexity and variability of the habitat of the cranes in Yancheng, there is a certain degree of subjectivity and uncertainty in the abovementioned research in determining the indicator weights. In respect to the study of the influencing factors of the habitat of the cranes, most of them were based on the influence of a single subsystem, and there was a certain degree of one-sidedness. Based on this, it would be worthwhile to improve and optimize the indicator assignment methods, and exploring the impact of measuring the habitat quality of the cranes with the EES indicator system will lead to a more comprehensive result. The following research delves further than the abovementioned research by focusing on the habitat of the red-crowned crane in Yancheng, constructing an evaluation index system of the habitat quality of the red-crowned crane based on the EES system, adopting the comprehensive "entropy weight method–analytic hierarchy process" evaluation model to assign weights for each index, and using the coupled coordination model to scientifically measure the interactions between the EES system and the quality of the habitat. Based on this foundation, we have analyzed the key factors influencing the quality of the habitat. To achieve this, we

combined LASSO regression optimization and the negative binomial regression model. Additionally, we utilized the support vector regression model improved by the random search algorithm to quantitatively assess the future population trends of the red-crowned cranes. The ultimate goal of our study is to propose scientific and effective measures that contribute to the conservation of the crane habitat and promote the animals' sustainable development. Simultaneously, it offers valuable practical insights for the conservation of nationally protected wild birds and the sustainable development of the ecological environment. The marginal contributions can be summarized into the following three points: First, we construct an evaluation index system for assessing the habitat quality of cranes, utilizing the EES system. This system takes into comprehensive consideration the influence of the social system, economic system, and ecosystem on the habitat quality of cranes. Second, we adopt the coupled coordination model to scientifically measure the degree of coordinated development within the EES system. Furthermore, we analyze the interaction between the EES system and the habitat quality of cranes. Third, we combine the LASSO regression and negative binomial regression models to analyze the crucial factors influencing the habitat quality of cranes. Subsequently, we conduct prediction analysis based on this analysis, enhancing the generalization ability and prediction accuracy of the model.

## 2. Materials and Methods

### 2.1. Study Area

The study focuses on the roosting area of overwintering red-crowned cranes in Yancheng, Jiangsu Province, which comprises five regions: Xiangshui, Binhai, Sheyang, Dafeng, and Dongta [5,15] (Figure 1). The Yancheng area of Jiangsu Province is situated in the northeastern part of Jiangsu Province (latitude 32°10′–34°21′ N, longitude 118°34′–121°08′ E). In terms of its national geographic location, Yancheng City is located in the coastal zone of eastern China and holds significant importance as part of the economic zones of eastern Shanghai, southern Jiangsu, and the coastal economic zone of Jiangsu. Yancheng is situated at the confluence of hills and plains, characterized predominantly by flat terrain; the southern region is encompassed by the Yangtze River Delta Plain, while the northern part of Jiangsu Province features the historic Yellow River Plain. The region exhibits minor undulations, with the highest elevation reaching up to 182 meters. The climate in the Yancheng area is a subtropical monsoon climate with an average annual temperature of about 13 °C and average annual rainfall of about 924 mm [16]. According to statistics, the population of red-crowned cranes in the area has shown a gradual increase since 1982, rising from about 200 to the current population of approximately 1000. As of 2001, a total of 1128 red-crowned cranes have been recorded in the Yancheng mudflat area. This number accounts for 94% of the total number of wintering cranes in China and 80% of the total number of migrating cranes in the world [6]. Due to the impact of human activities on the climate and environment, the habitat of cranes is constantly facing destruction and significant fragmentation. Consequently, the population of cranes has experienced a decline in recent years [17]. Therefore, conducting scientific measurement and an analysis of its key influencing factors, as well as further exploring strategies, is necessary for the preservation of cranes and their habitat.

### 2.2. Indicator Selection

To scientifically measure the interaction between the local EES system and the carne habitat in Yancheng, as well as analyze the factors influencing habitat quality, this study combines relevant studies by Peng Huang and Jingjing Dang [18,19]. The ecosystem indicators selected for analysis include the normalized vegetation index, regional annual rainfall, and soil erosion area. For the economic system, indicators include the total economic volume (including regional GDP, total food output, and local fiscal revenue), economic quality (growth value of the three major industries, per capita disposable income), and economic structure (the proportion of the three major industries). Indicators related to social system include the demographic situation (natural growth rate, population density),

social development, and people's lives (Engel coefficient, public library collection, and employment structure). The relevant variables are presented in Table 1.

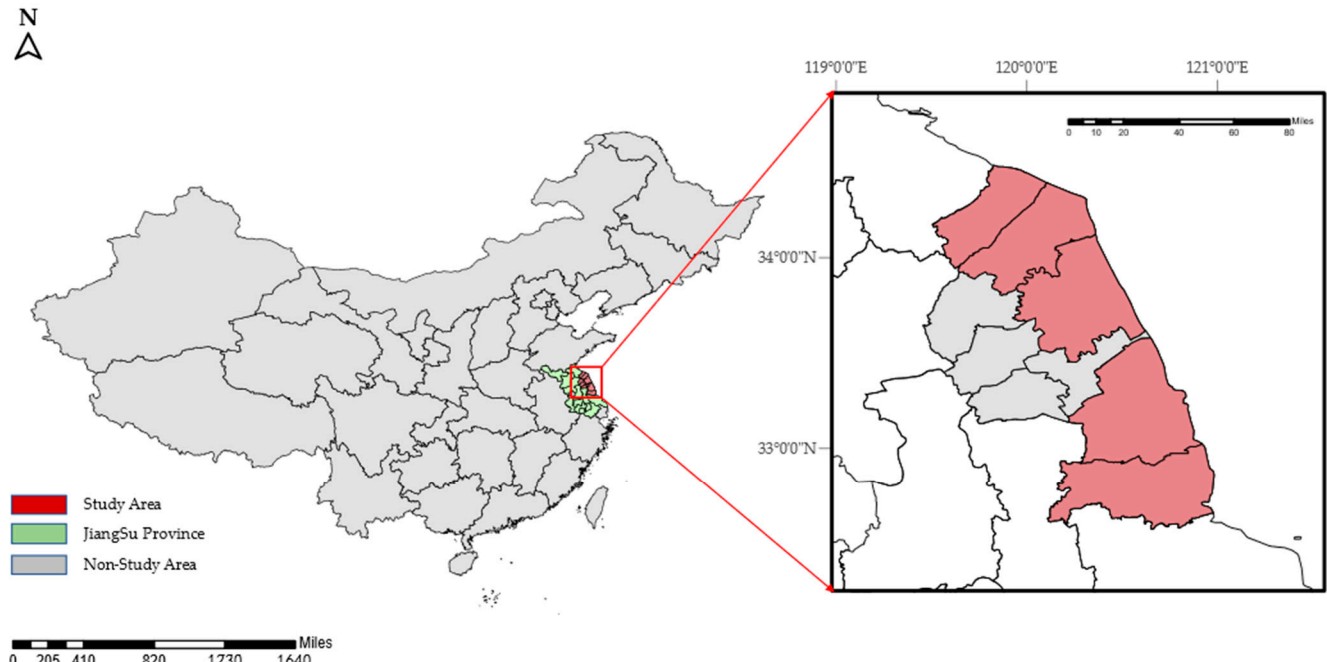

**Figure 1.** Map of the study area (red area) and the province to which the study area belongs (green area). The geographical location of the study area is identified by a red box on the map of China Map.

**Table 1.** Evaluation indicators system of EES system in Yancheng, Jiangsu Province.

| Target Layer | Dimension Layer | Indicator Layer |
|---|---|---|
| EES System in Yancheng, Jiangsu Province | Ecological system | Normalized difference vegetation index (NDVI) <br> Annual precipitation (AP) <br> Soil erosion area (SEA) |
| | Economic system | Regional GDP (RGDP) <br> Total food output (TFO) <br> Local fiscal revenue (LFR) <br> Growth of the primary industry (GPI) <br> Growth of the secondary industry (GSI) <br> Growth of the tertiary industry (GTI) <br> Per capita disposable income (PCDI) <br> Proportion of primary industry (PPI) <br> Proportion of secondary industry (PSI) <br> Proportion of tertiary industry (PTI) |
| | Social system | Natural growth rate (NGR) <br> Population density (PD) <br> Rural Engel coefficient (REC) <br> Urban Engel coefficient (UEC) <br> Public library collection (PLC) <br> Employees in primary industry (EPI) <br> Employees in secondary industry (ESI) <br> Employees in tertiary industry (ETI) |

*2.3. Data Source and Pre-Processing*

To obtain data for the above indicators, multiple data sources were used in this study, including national statistical agencies, publicly available government data, academic databases, and research results. The data used in this study comprise indicators in the

domains of national economy, ecology and environment, and demographics. Specifically, economic indicators were sourced from the National Bureau of Statistics of China (NBSC) and the China Economic and Social Development Statistics Database (CESD), which encompass variables such as gross regional product, value added and share of each industry, local fiscal revenue, total food production, per capita disposable income, Engel's coefficient, employment structure, and more. Data on demographic indicators were derived from the information on natural growth rate and population density released by the NBSC; data pertaining to culture and education were obtained from the library collection indicators of the China Economic and Social Development Statistics Database. Moreover, the analysis of the ecosystem in the Yancheng area incorporates indicators such as annual rainfall, sourced from the statistical yearbook of the region. Additionally, the normalized difference vegetation index (NDVI) was obtained from a study in the literature titled "The spatio-temporal evolution characteristics and response of regional climate change of NDVI at Jiangsu coastal areas" [20]. For soil erosion areas, we used the data provided by the China Water Resources Bureau. In analyzing the number and distribution of red-crowned cranes in the Yancheng, we referenced a study from the literature titled "Dynamic changes in population size and habitat distribution of wintering red-crowned cranes in northern Jiangsu Province" [21] as the source of our data. It is worth noting that these data sources were rigorously screened, and quality control measures were implemented in this research to ensure the reliability and accuracy of the data. These efforts provide reliable data support for the analysis conducted in this study.

Furthermore, this section aims to select a suitable method for filling in missing data. Upon investigating the missing data, it can be observed that the percentage of missing values among the 21 indicators is distributed between 0% and 45% (Figure 2a). The Spearman correlation coefficient heat map among the indicators reflects a certain correlation among the missing data (Figure 2b), which can be classified as a random missing type. Based on the above exploration of missing data types, using sample statistics to fill in the data would reinforce the sample bias in the original data and result in model distortion. Therefore, in this section, a multiple imputation dataset was constructed based on the multiple imputation by chained equations (MICE) method [22,23]. The predictive mean matching (PMM) method algorithm was used to calculate the initial filled values, and multiple regression models were constructed to fill in the missing values using data from other observed columns after several iterations until convergence [24].

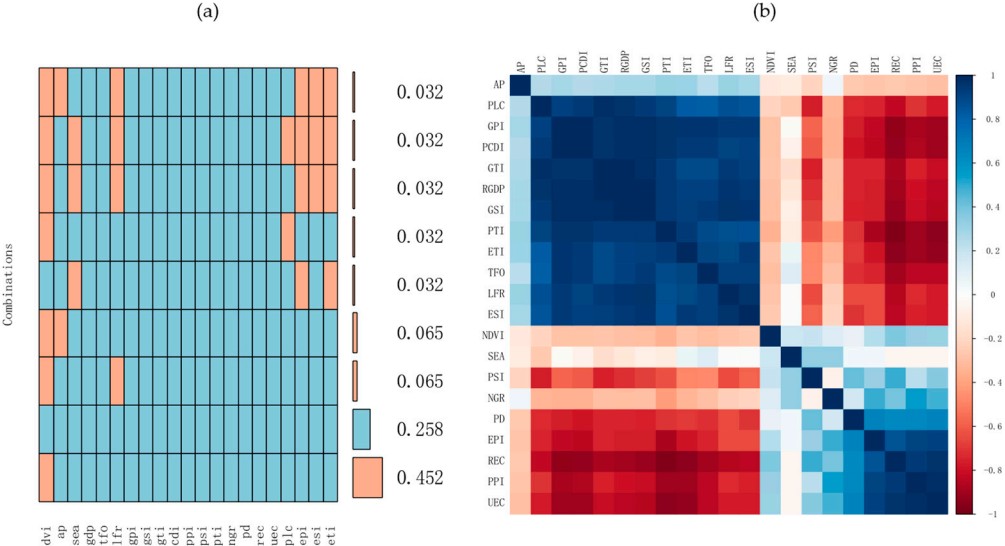

(a)                                                    (b)

**Figure 2.** (**a**) Missing Data Model. (**b**) The spearman rank correlation coefficient heat map.

*2.4. "EWM-AHP" Comprehensive Evaluation Model*

This study used a comprehensive evaluation model based on the entropy weight method (EWM) and analytic hierarchy process (AHP) to assess and assign weights to the indicators of the EES composite system, covering the period from 1990 to 2020. The main reason for utilizing the aforementioned method is based on the following two considerations: First, considering the large number of indicators involved in this study, it is impractical to invite experts from various fields to assess the importance of each indicator due to resource constraints. Solely relying on a subjective assignment method would lead to bias in measuring the degree of coordinated development within the EES system. Second, a large portion of the indicators in the study have missing data. Only using the objective weighting method would significantly increase the error between the interpolated data and the observed values. However, using the comprehensive "EWM-AHP" evaluation model can effectively improve the accuracy of the subsequent model's results. Third, considering the differences in the influence of the evaluation indexes within each subsystem on the EES system, the allocation of scientific and reasonable weight will directly determine the accuracy of subsequent model analysis. The specific processes of the model's construction are detailed in the following section.

2.4.1. Entropy Weight Method

The entropy weighting method is an objective assignment method. For a certain indicator, the entropy value can be used to assess its dispersion degree. A smaller entropy value implies a higher level of dispersion for the indicator and a greater influence on the comprehensive evaluation. If all values of a certain indicator are equal, the indicator does not contribute to the comprehensive evaluation [25]. The main steps are as follows:

The first step is to construct the initial sample matrix $X = (x_{ij})_{n \times m}$. To eliminate the effect of indicator data size, variability, and magnitude of values, normalization is performed using Equation (1) [26]:

$$x'_{\square j} = \begin{cases} \frac{x_{ij} - \min\{x_j\}}{\max\{x_j\} - \min\{x_j\}}, & \text{Positive Indicators} \\ \frac{\max\{x_j\} - x_{ij}}{\max\{x_j\} - \min\{x_j\}}, & \text{Negative indicators} \end{cases} \tag{1}$$

In the second step, the information entropy of each indicator is determined by using Equation (2) [27]:

$$e_j = -\frac{1}{\ln m} \sum_{i=1}^{m} y_{ij} \times \ln y_{ij} \tag{2}$$

In Equation (2), $e_j$ represents the information entropy of the indicator $j$. When $x_{ij} = 0, y_{ij} = 0, y_{ij} \times \ln y_{ij}$ is meaningless; therefore, the corrected specific gravity equation is:

$$y_{ij} = \frac{x_{ij}' + 10^{-4}}{\sum_{i=1}^{m}(x_{ij}' + 10^{-4})} \tag{3}$$

In the third step, the entropy weight of each indicator is determined by using Equation (4):

$$w_j = (1 - e_j) / \sum_{i=1}^{m}(1 - e_j) \tag{4}$$

In Equation (4), $w_j$ is the entropy weight of the indicator $j$, where $w_j \in [0, 1]$ and $\sum_{i=1}^{n} w_j = 1$.

2.4.2. Analytic Hierarchy Process

The analytic hierarchical process [28,29] decomposes the problem into different constituent factors based on its nature and the overall objective. It then aggregates and

combines these factors at different levels according to their interrelated influences and affiliations. This process forms a multi-level analysis structure model, ultimately reducing the problem by determining the relative importance of the weights of the lowest level relative to the highest level or the ranking of the relative advantages and disadvantages. The main steps are as follows:

In the first step, the initial judgment matrix $P = (r_{ij})_{m \times n}$ is constructed based on the comparison of the scalar assignments [30] (Table 2), where $r_{ij}$ denotes the evaluation index $j$ of the sample $i$, and $r_{ji} = 1/r_{ij}$.

**Table 2.** Comparison scales.

| Standard Value | Definition | Description |
|:---:|:---|:---|
| 1 | Equally important | Factors $r_i$ and $r_j$ are of equal importance |
| 3 | Slightly important | Factors $r_i$ and $r_j$ are slightly more important |
| 5 | Comparatively important | Factors $r_i$ and $r_j$ are of high importance |
| 7 | Obviously important | Factors $r_i$ and $r_j$ are significantly more important |
| 9 | Absolutely important | Factors $r_i$ and $r_j$ are absolutely high in importance |

In the second step, according to the positive definite reciprocal inverse matrix $P$, its maximum characteristic root $\lambda_{\max}$ and the weight of each indicator are determined using Equations (5) and (6) [31]. We normalize each column of the judgment matrix and then sum the normalized judgment matrix by rows. The vector $w = \left( \bar{w}_1, \bar{w}_2, \ldots, \bar{w}_n \right)^T$ is obtained by normalizing this sum:

$$w_i = \frac{w_i}{\sum_{i=1}^{n} w_i} (i = 1, 2 \ldots, n) \tag{5}$$

$$\lambda_{\max} = \sum_{i=1}^{n} \frac{(PW)_i}{nW_i} \tag{6}$$

In Equation (5), $w_i = \sum_{j=1}^{m} \frac{r_{ij}}{\sum_{i=1}^{n} r_{ij}} (i, j = 1, 2, \ldots, n)$.

The third step is the consistency test. In practice, there is a deviation between the values of $\lambda_{\max}$ and $n$ due to the specific composition of the judgment matrix in the analysis hierarchical process. To avoid excessive errors affecting the judgment results, it is essential to test the consistency of the judgment matrix. The formula for the consistency index is defined as follows:

$$CR = \frac{CI}{RI} = \frac{\lambda_{\max} - n}{(n-1)} \tag{7}$$

In Equation (7), $n$ represents the order of the pairwise comparison factor, $CI$ denotes the consistency test indicator, and $RI$ represents the average random consistency index [32], which takes different values depending on the order of the pairwise comparison factor. $CR$ is the consistency ratio of the judgment matrix. If $CR < 0.1$, it is considered that the inconsistency degree of $P$ is within the tolerance range, indicating satisfactory consistency, and the matrix passes the consistency test. Otherwise, the comparison matrix $P$ should be adjusted again.

### 2.4.3. Integration Weights Based on the Dispersion Function

To make full use of the objectivity of decision information and take into account the decision maker's perception of habitat indicators, this study combines the subjective weight $u_i$ and the objective weight $v_i$. It introduces the deviation function [33], establishes the objective planning model, and sets the preference factor $\mu = 0.5$ for the deviation function, which ensures that the subjective and objective weights have the same degree of influence on the combined weights $W = (w_1, w_2, \ldots, w_m)^T$. The main steps are as follows:

In the first step, a deviation function is constructed to quantify the degree of deviation in the subjective and objective weights relative to the combined weights:

$$_i = \sum_{j=1}^{m} \left[ (w_j - u_j) z_{ij} \right]^2 \tag{8}$$

$$h_i = \sum_{j=1}^{m} \left[ (w_j - v_j) z_{ij} \right]^2 \tag{9}$$

In Equations (8) and (9), $d_i$ denotes the deviation in the decisions made by the subjective assignment method (analytic hierarchy process) compared to the integrated weights, $h_i$ denotes the deviation in the decision made by the objective assignment method (entropy weight method) compared to the integrated weights, and $z_{ij}$ is the normalized processed data.

In the second step, an objective planning model is constructed by Equation (10) [33], aiming to minimize the total deviation:

$$min \ \mu \sum_{i=1}^{n} {}_i + (1 - \mu) \sum_{i=1}^{n} h_i$$
$$s.t \ \sum_{j=1}^{m} w_j = 1, w_j \geq 0 \tag{10}$$

In the third step, the Lagrangian function is introduced, and the planning model is solved. The Lagrangian function incorporates the constraints and objectives of the model and allows for optimization. By solving the Lagrangian function, the optimal solution for the planning model can be obtained:

$$L(w, \lambda) = \mu \sum_{i=1}^{n} \sum_{j=1}^{m} \left[ (w_j - u_j) z_{ij} \right]^2 + (1 - \mu) \sum_{i=1}^{n} \sum_{j=1}^{m} \left[ (w_j - v_j) z_{ij} \right]^2$$
$$+ 2\lambda \left( \sum_{j=1}^{m} w_j - 1 \right) \tag{11}$$

*2.5. Coupling Coordination Development Model*

The coupled coordination degree development model is a research method developed for complex social and economic systems. It explores the interactions between multiple subsystems to comprehend the dynamic evolution of the whole system. The model takes an integrated perspective to describe and study each subsystem, taking into account their interdependencies and interactions. In this study, this method is adopted to investigate the coordinated development degree of the EES system based on the following considerations: First, the fuzzy affiliation function model [34] can only calculate the coordination degree between two subsystems, while the coupled degree coordinated development model [35] can evaluate the coupled coordination degree of multiple subsystems. In this study, there are three subsystems, ecological, economic, and social, which are suitable for the coupled coordinated development model. Second, traditional analysis methods usually regard the system as an independent subsystem, ignoring the interactions between subsystems. The coupled and coordinated development model allows a global perspective analysis to take place, examining the interactions and impacts of different subsystems within the system. It enables a comprehensive analysis of the coupling relationship among all subsystems and the system as a whole. The specific model formulations are as follows.

2.5.1. Coupling Degree Model

For the coupling degree, as an important indicator to reflect the degree of coupling among regional ecological, economic, and social systems, it is important to discern the intensity and extent of interaction within EES systems. It enables the quantification and

analysis of the development trend of EES systems. The coupling degree implies the correlation and closeness between subsystems, and its algorithm is as follows:

$$C = \sqrt{\left[1 - \frac{\sum_{i>j,j=1}^{n} \sqrt{(U_i - U_j)^2}}{\sum_{m=1}^{n-1} m}\right] \times \left(\prod_{i=1}^{n} \frac{U_i}{maxU_i}\right)^{\frac{1}{n-1}}} \tag{12}$$

In Equation (12), $U_i$ represents the value of the subsystem $i$, where $U_i \in [0,1], C \in [0,1]$. The EES system under study in this paper is a dynamic system, and the coordination degree $C_1, C_2, \ldots, C_{26}$ of the system is calculated from the above equation year by year. A higher $C$ value indicates a lower level of discrepancy between subsystems and a higher the coupling degree. Conversely, a higher level of discrepancy between subsystems leads to a lower coupling degree.

2.5.2. Comprehensive Evaluation Index

The comprehensive evaluation index is an indicator that reflects the comprehensive performance level of the evaluated object across various aspects by combining multiple indicators with certain weights. By using the comprehensive evaluation index, the performance of the EES system can be compared and evaluated in multiple aspects, providing a comprehensive understanding of its overall level and identifying its strengths and weaknesses. The algorithm for calculating the comprehensive evaluation index is as follows:

$$T_i = \sum_{j=1}^{m} \sum_{i=1}^{n} \alpha_j \times x_{ij}', \sum_{j=1}^{m} \alpha_j = 1 \tag{13}$$

In Equation (13), $T_i$ represents the composite evaluation index for the year $1989 + i$, $x_{ij}'$ is the standardized value of the indicator $j$ in the year $1989 + i$, and $\alpha_j$ denotes the weight of the indicator $j$.

2.5.3. Coordinated Development Degree

The coupled coordinated development model establishes connections and interactions among multiple subsystems to form a whole system. By examining the interactions and influences among different subsystems, it explores the evolutionary patterns and trend of "coordinated development" within the whole system. The algorithm for this model is as follows:

$$D = \sqrt{C \times T} \tag{14}$$

*2.6. LASSO Regression Model*

The least absolute shrinkage and selection operator (LASSO) regression model [36] is a method used for feature selection and sparse modeling. Its objective is to reduce model complexity and generalization errors via regularization. Compared with traditional variable selection methods, this study uses LASSO regression for a variable selection based on the following considerations: First, LASSO regression performs better in high-dimensional datasets, allowing the selection of variables that contribute more to the response variable to take place. This avoids the limitations of methods such as stepwise regression in high-dimensional datasets. Additionally, the penalty term of LASSO regression can compress the coefficients of certain variables to zero, effectively selecting variables with significant effects on the response variables. This enhances the predictive performance of the model. Second, LASSO regression controls complexity through the penalty term, mitigating the issue of overfitting. It can achieve satisfactory results even with small sample sizes, considering multiple relevant variables simultaneously. In contrast, stepwise regression generally considers only one variable at a time and requires additional techniques such as cross-

validation to avoid overfitting. The optimization problem of LASSO regression can be formulated as follows:

$$min \, RSS + \lambda \Sigma |\beta_j| \tag{15}$$

In Equation (15), $RSS$ denotes the residual sum of squares, where $\beta_j$ is the coefficient of the predictor variable $j$, and $\lambda$ is a regularization parameter that controls the strength of the penalty. As $\lambda$ increases, the value of many $\beta_j$ will shrink until finally some coefficients become zero.

### 2.7. RS-SVR Model

Support vector regression (SVR) [37] is an application scenario of support vector machines (SVM) to regression problems. In this study, the SVR model is used to predict the population size of red-crowned crane and analyze the evolutionary dynamics of their habitat based on two main considerations: First, traditional linear regression is a strict regression model with the loss of the combined distance from the actual location of all samples to this linear function. The parameters of the linear function are estimated by minimizing the loss. Linear regression models tend to overfit as any sample deviating from the linear function contributes to the loss. Second, for datasets with a limited sample, the SVR model demonstrates high validity by minimizing the object function and inferring relationship over the entire datasets. In the SVR model, the least squares method is usually used as the loss function [38], with:

$$min_{\omega, b, \xi_i, \hat{\xi}_i} \frac{1}{2} \left\| w^2 \right\| + C \sum_{i=1}^{m} (\xi_i, \hat{\xi}_i) \tag{16}$$

$$s.t. \begin{cases} f(x_i) - y_i \leq \varepsilon + \xi_i \\ y_i - f(x_i) \leq \varepsilon + \hat{\xi}_i \\ \xi_i \geq 0, \hat{\xi}_i \geq 0, i = 1, 2, \ldots, m \end{cases} \tag{17}$$

In Equations (16) and (17), $\xi_i$ and $\hat{\xi}_i$ represent the relaxation variables, $C$ denotes the regularization parameter, and $\varepsilon$ is the tolerance. In the SVM model, a prediction error is considered intolerable when the distance between the predicted value and actual value exceeds the tolerance $\varepsilon$. The loss function in SVM is minimized when $f(x)$ exactly matches $y$. However, in support vector regression, it assumes that a deviation of up to $\varepsilon$ between $f(x)$ and $y$ can be tolerated. Consequently, the loss is only calculated when the absolute difference between $f(x)$ and $y$ exceeds $\varepsilon$. In this case, an interval band of width $2\varepsilon$ is constructed with $f(x)$ as the center. If the training samples fall within this interval band, they are considered correctly predicted.

For parameter tuning in SVR models, optimization methods such as the grid search method are commonly used to optimize the model by exploring different combinations. The main parameters of the SVR model are as follows:

$C$: The regularization parameter controls the trade-off between minimizing the error and controlling the complexity of the model. A larger value of C imposes a stronger penalty of the error, leading to a more precise fit to the training data.

$\gamma$: The width parameter of the radial basis function determines the influence of individual data points on the model's output. It controls the size of the area within which data points have an impact on the prediction.

In this study, the random searching (RS) algorithm [39] is used to optimize the regularization parameter and width parameter in the SVR model. The RS algorithm can be characterized as an optimization technique that combines both local and global search strategies with random sampling. Its optimization problem can be expressed as:

$$min f(x), x \in X \tag{18}$$

In Equation (18), $f(x)$ represents the objective function, $X$ denotes the definition domain, and the objective is to find a locally optimal solution or a globally optimal solution

within $X$. The random search algorithm operates by initial solution $x_0$ and then iteratively updates the solution using random perturbations. At each iteration, a better solution $t_i$ is generated, satisfying the condition $F(x_i + 1) \leq F(x_i)$, and this process continues until either the global optimal solution is found or a specified stopping criterion is reached.

Using all available data for training a model can lead to overfitting, where the model becomes too specific to the training data and performs poorly on unseen data. To avoid this issue, K-FOLD cross-validation is commonly used. It involves dividing the dataset into K subsets, using each subset in turn as the test set while using the remaining K-1 subsets as the training set. This process is repeated K times, allowing for a comprehensive assessment of the model's performance within all the available data. Additionally, K-FOLD cross-validation enables the SVR algorithm to generalize better to unknown datasets [40]. The leave-one-out method is a special case of K-FOLD cross-validation, where the number of training samples is used as the number of cross-folds. Previous studies have shown that the leave-one-out method (LOOCV) [41,42] consistently approaches optimal results, particularly when the size of the training samples is limited. Thus, in this study, we use the leave-one-out method for cross-validate models.

## 3. Results

### 3.1. Index System and Coordinated Development Degree Analysis of EES System

Previous studies have demonstrated a certain relationship between the conservation of crane habitat, regional economic development, and the ecological environment in the coastal area of Yancheng [10,11]. However, according to the weighting results (Figure 3), the entropy weighting method obtained a weighting of 78.04% for the social system, 12.17% for the ecosystem, and 9.79% for the economic system, while the analytic hierarchy process obtained a weighting of 56.95% for the economic system, 33.31% for the social system, and 9.74% for the ecosystem. Consequently, by using the comprehensive "EWM-AHP" evaluation model, the social system was assigned a weight of 41.71%, the economic system was assigned a weight of 29.35%, and the ecosystem was assigned a weight of 28.94%.

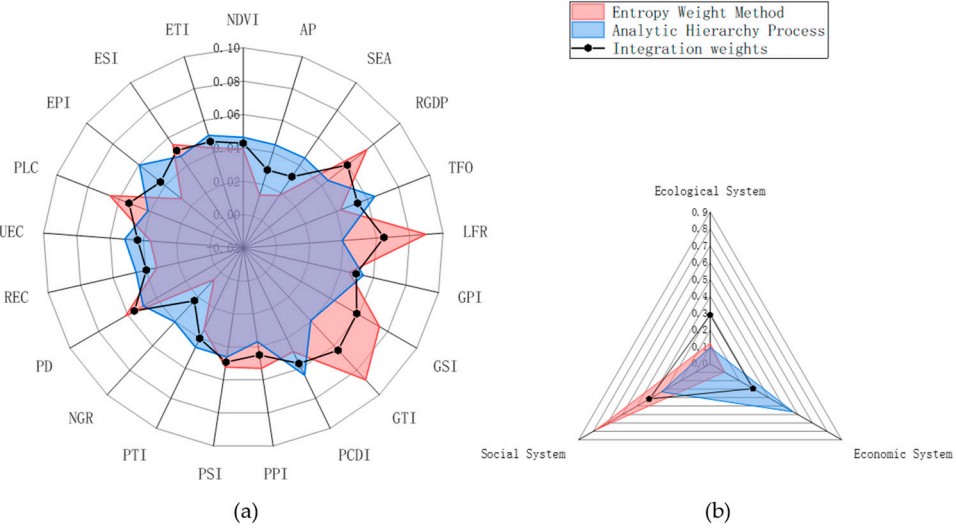

**Figure 3.** (**a**) Results of EES system index comprehensive weights. (**b**) Results of comprehensive weights of economic, ecological and social systems.

Furthermore, the application of the coupled coordination degree development model provided a scientific measurement of the degree of coordinated development within the EES system. As illustrated in Figure 4, the population size of the cranes exhibited fluctuations from 1990 to 2015, with the highest population size recorded as 1128. Correspondingly, the degree of coordinated development of the EES system also demonstrated similar fluctuations, which indicated a close relationship between the quality of the crane habitat

and the degree of coordinated development within the EES system. The verification of their correlation can be obtained with a Spearman correlation coefficient of 0.69, indicating a significant interaction between the two factors. Additionally, the degree of coordinated development within the EES system showed an overall increasing phenomenon from 0.2236 in 1990 to 0.4175 in 2015.

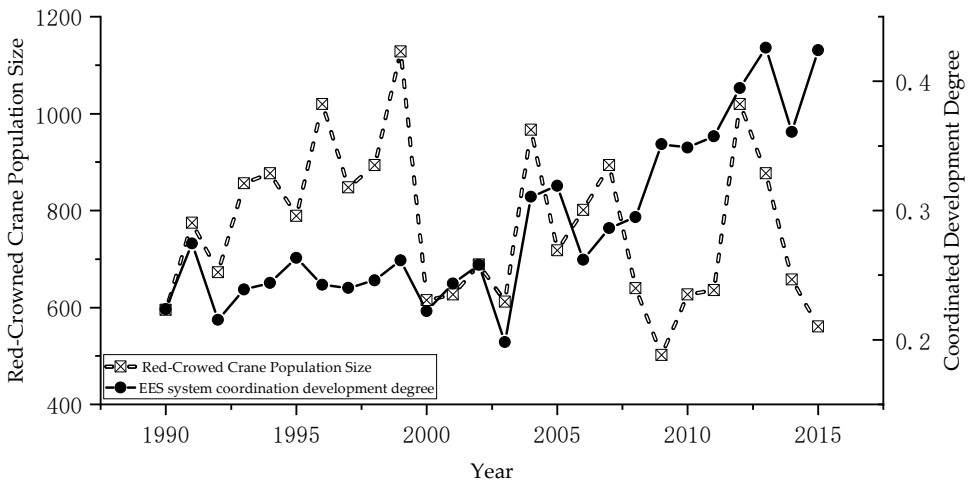

**Figure 4.** Coordinated development degree of EES system and population trend of red-crowned.

However, it is noteworthy that in 1997, 2009, and 2013, the population size of red-crowned crane exhibited an opposite trend with the degree of coordinated development within the EES system. Surprisingly, the population size showed a significant decline despite the increasing trend in the system's degree of coordinated development, with decrease rates of 16.86%, 21.56%, and 14.02% accordingly. Moreover, although the overall degree of coordinated development within the EES system showed an increase with a growth rate of 61.35%, it is important to highlight that the highest degree of development achieved during the sample period was only 0.4175 from a numerical point of view, and there is still much potential for improvement.

*3.2. Influencing Factors and Prediction of Habitat Quality of the Red-Crowned Crane*

3.2.1. Influencing Factors of Habitat Quality of the Red-Crowned Crane

Due to the high dimensionality of the indicators in the original data, an important consideration was to reduce the dimensionality of all variables by using principal component analysis [43]. The results showed (Table 3) that the first three principal components accounted for 84.85% of the explained variance. However, the explanations of the first three principal components were not sufficiently comprehensive, so further reduction in the dimensionality of some of the indicators was considered. A visual analysis of Figure 2b revealed a strong correlation between the growth value, the number of employees, and the proportion of the three major industries. Consequently, principal component analyses were conducted for these indicators, leading to the following findings (Table 3): The variance of the first principal component of the growth value of the three major industries is 98.57%, which can be explained as the "comprehensive industrial growth value". Similarly, the variance of the first principal component of the number of employees in the three major industries is 85.85%, which can be explained as "urbanized employment structure". Additionally, due to the sum of the three major industries being equal to one, the variance of the first two principal components already achieved 100%.

**Table 3.** Percentage of cumulative variance explained by principal component analysis.

| Variable | PCA1 | PCA2 | PCA3 |
|---|---|---|---|
| Total variables | 70.16% | 77.84% | 84.85% |
| GPI; GSI; GTI | 98.57% | 99.81% | 100% |
| EPI; ESI; ETI | 85.85% | 98.51% | 100% |
| PPI; PSI; PTI | 73.83% | 100% | 100% |

In this study, we fit a generalized linear model (GLM) based on the Poisson family and negative binomial family of the above explanatory variables to deeply analyze the factors affecting the habitat quality in the Yancheng area. Due to the high dimensionality of the variable set, the fitted GLM model exhibits a significant degree of multicollinearity. To eliminate this phenomenon, this study conducted LASSO regression to select significant explanatory variables from the variable sets. The results revealed that the best combined results, considering both residual squared and penalty terms, retained six variables (Figure 5a): normalized vegetation index, annual rainfall, soil erosion area, natural population growth rate, Engel's coefficient of rural residents, and public library collection. Subsequently, the GLM models based on Poisson and negative binomial families were fitted using these six variables. Notably, the negative binomial regression model optimized based on LASSO regression exhibited the lowest AIC and BIC values (Table 4), which are 396.58 and 408.05, respectively. Further analysis indicated that the residuals of the model were uniformly distributed and adhered to the Gaussian–Markov assumption (Figure 5b,d). Additionally, the model passed the normality test (Figure 5c), indicating minimal influence from outliers. However, a few influential points were identified, which affected the fitting effect (Figure 5e).

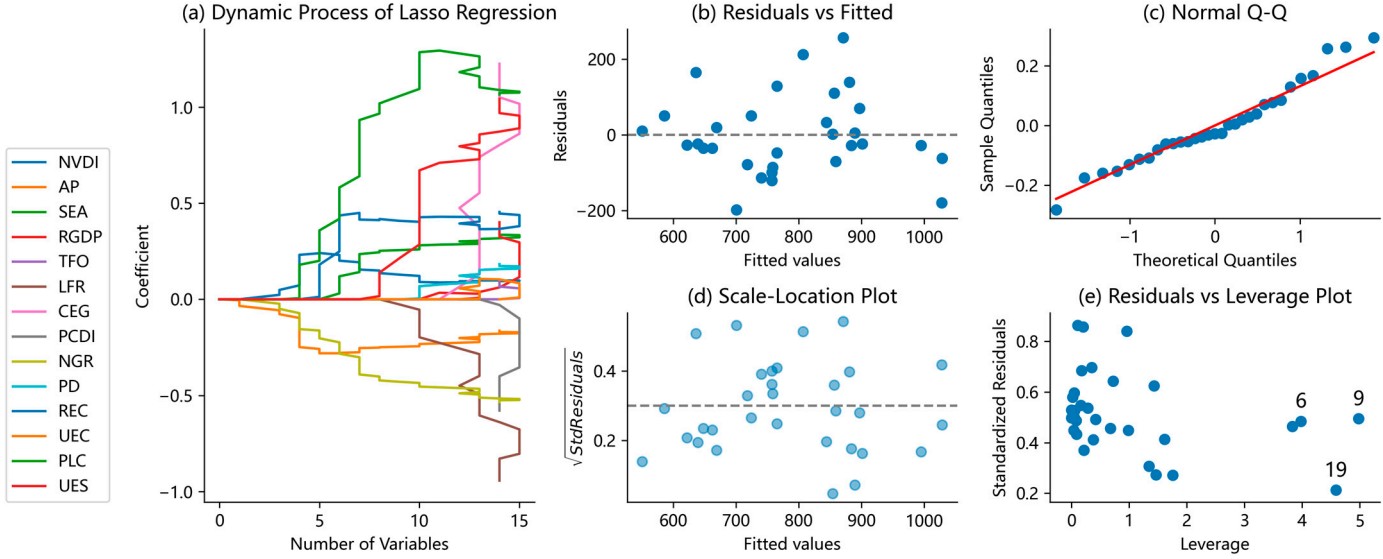

**Figure 5.** (**a**) Variable selection process for LASSO regression. (**b**) LASSO-negative binomial regression model residual test. (**c**) LASSO-negative binomial regression model normality test. (**d**) LASSO-negative binomial regression model residual distribution test. (**e**) LASSO-negative binomial regression model outlier test.

### 3.2.2. Prediction of Habitat Quality of the Red-Crowned Crane

To gain a better understanding of the factors influencing the habitat quality of the cranes, the SVR model was used to predict and analyze the population size of the cranes in the Yancheng area. The SVR model was constructed by combining the index weights obtained from the comprehensive "EWM-AHP" evaluation model with various optimization algorithms (Figure 6). The results revealed that the cross-validation scores of the original

model (SVR), the genetic algorithm-based model (GA-SVR), and the Bayesian-optimization-algorithm-based model (BO-SVR) ranged from −0.4401 to −0.2364. Furthermore, the training set scores of the GA-SVR and BO-SVR models are 0.9820 and 0.9537, respectively. These scores indicate that the GA-SVR model and BO-SV model are overfitted and exhibit a lower prediction accuracy. In contrast, the model based on the random search algorithm (RS-SVR) demonstrated improvements in both the training set score and the cross-validation score compared to the original model, with 9.39% and 52.64%, respectively.

**Table 4.** Model comparison.

| Model | AIC | BIC |
|---|---|---|
| Poisson regression | 756.04 | 777.55 |
| Negative binomial regression | 409.92 | 432.87 |
| LASSO-Poisson regression | 783.09 | 772.89 |
| LASSO-negative binomial regression | 396.58 | 408.05 |

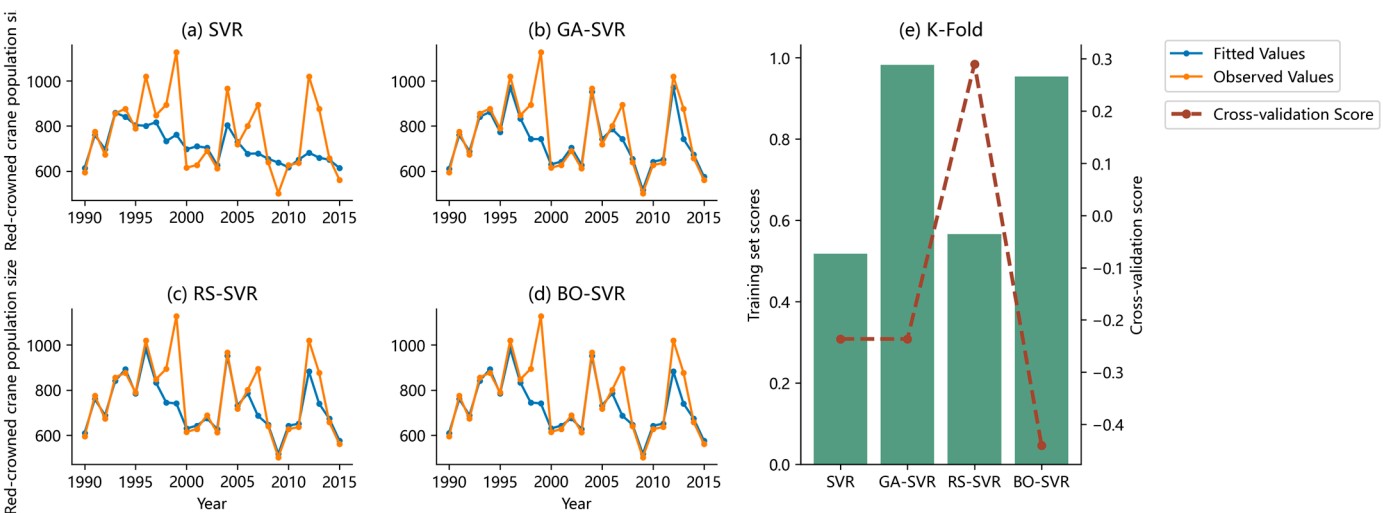

**Figure 6.** (**a**) Fitted graph of SVR model with default parameters. (**b**) Fitted graph of SVR model based on genetic algorithm optimization. (**c**) Fitted graph of SVR model optimized based on random search algorithm. (**d**) Fitted graph of SVR model optimized based on Bayesian algorithm. (**e**) Comparison of cross-validation results.

Based on the predicted results of the RS-SVR model for the population size of the red-crowned crane in the Yancheng area from 2016 to 2020 (Figure 7), it can be observed that the population size of the red-crowned crane was 877 in 1994, while it decreased to 561 in 2015. This decline shows a fluctuating downward phenomenon with an overall decline rate of about 36.03% during the sample period. Furthermore, according to the predicted data of the model, since 2015, the population size of the cranes has been on an upward movement with a growth rate of about 69.70%.

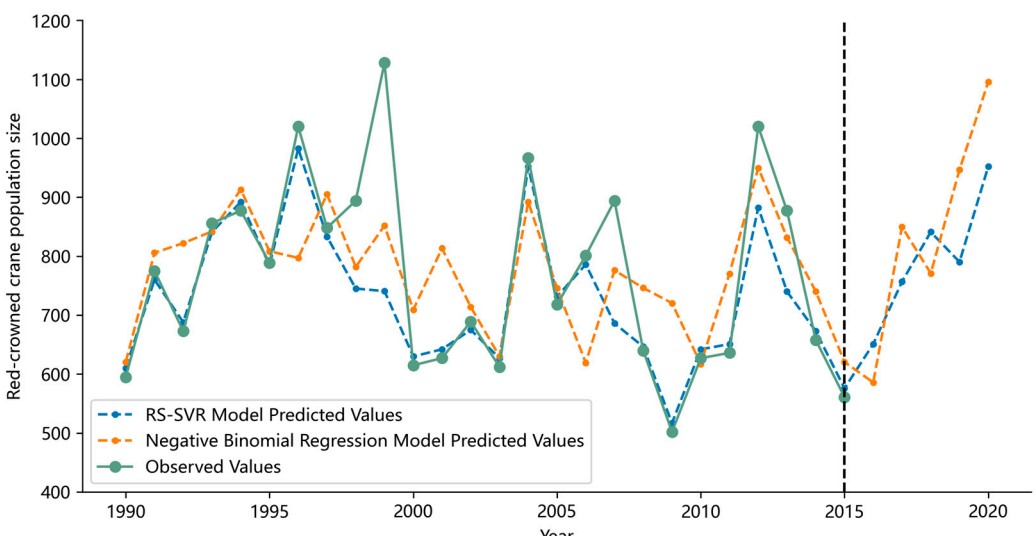

**Figure 7.** Predicted population size of red-crowned cranes in Yancheng, Jiangsu.

## 4. Discussion

### 4.1. Degree of Coordinated Development of EES Systems

The results of the EES system indicator weights reveal that the social system has a more significant impact on the habitat quality of the crane compared to the ecological and economic systems. This suggests that factors such as excessive population growth and extensive human activities can lead to the reduction in and destruction of land resources. In addition, these factors can also lead to habitat damage and fragmentation, which is consistent with previous studies [44]. In addition, changes in agricultural activities can affect the relationship between residents and the local ecological environment. Intensity agricultural production and the excessive use of pesticides can lead to further pollution and destruction of the habitat in the Yancheng area.

The degree of the coordinated development of the EES system is an important indicator to identify the relationship and intensity of the interaction among the local ecological, economic, and social systems. It reflects the degree of coordinated development. The close interaction between the EES and population size of the crane reflects that the protection of the ecological environment is a necessary way to maintain ecological balance and promote coordinated development. At the same time, the impact of human social and economic activities on the natural environment can also lead to ecosystem destruction and species extinction. Therefore, improving the degree of coordinated development within the EES system is conducive for protecting the ecological environment, thus promoting the survival of endangered species and the breeding of their habitats.

It is worth noting that in 1997, 2009, and 2013, the population size of the cranes exhibited opposite trends to the degree of coordinated development within the EES system in the Yancheng area. These discrepancies can be attributed to several factors:

- In 1997, sudden heavy rainfall in the Yancheng area caused severe flooding. This series of factors led to serious damage to the local ecological environment, including habitat fragmentation, reduced food resources, etc.
- In 2009, unusual snowfall occurred in Yancheng, prompting a large number of cranes to leave the area in search of food [8,11]. At the same time, snowfall also froze large areas of water and wetlands, depriving the cranes of sufficient food and water and making them susceptible to disease. These extreme conditions resulted in severe impacts on both the cranes' habitat and its wintering grounds.
- In 2013, activities such as local industrial expansion and agricultural polder in Yancheng exerted considerable pressure on the cranes' habitat [7].

Based on the above results, the relevant policy recommendations are as follows:

1.  It is crucial to strengthen coordinated development planning. When formulating local economic and social development plans, wildlife protection must be considered. Measures such as control engineering, transportation networks, policies, and regulations should be adopted to reduce the damage caused by human activities to wildlife habitats.

2.  There is a need to prioritize the reasonable layout of protected areas. In the process of regional planning, wildlife protection areas should be reasonably planned to effectively separate wildlife habitats from economic development areas. Implementing physical isolation measures can help protect wildlife habitats so as to achieve a harmonious balance between economic growth and conservation objectives.

3.  To protect wildlife habitats, the management and monitoring of wildlife should be strengthened. This involves establishing effective ecological protection mechanisms, formulating strict laws and regulations on wildlife protection, and improving the construction and management of protected areas. These measures are necessary to ensure that the quality of wildlife habitats is not negatively affected by human activities.

4.  Promoting sustainable development practices is vital. When pursuing economic development, it is crucial to adopt a sustainable industrial development model and utilize natural resources in a responsible manner. Investment in ecological and environmental protection infrastructure should be prioritized to reduce the negative impact on wildlife habitats and enhance their long-term stability.

### 4.2. Key Habitat Influencing Factors

The population size of local red-crowned cranes is considered the response variable in this study as it provides a direct indication of the crane habitat's quality. This variable represents descriptive count type data, where the mean is smaller than the variance in the sample range. To analyze these data, generalized linear regression models based on Poisson distribution, negative binomial distribution, and probability distributions are fitted. The optimal model is selected by comparing the AIC and BIC values of different models. The results reveal that several factors significantly impact the population size of red-crowned cranes, including the normalized vegetation index, annual precipitation, soil erosion area, natural population growth rate, Engel's coefficient of rural residents, and public library collection. These findings align with previous studies conducted in Yancheng [45]. Specifically, the factors which directly or potentially affect the quality of the habitat of the red-crowned crane are as follows:

The normalized difference vegetation index (NDVI) plays an important role in quantifying the density and overall health of vegetation within an ecosystem. It provides a reliable measure of vegetation cover, and higher NDVI values indicate improved vegetation abundance and conditions. In the context of red-crowned crane habitats, the presence of dense vegetation is extremely important. Dense vegetation not only provides adequate food resources for cranes but also provides critical shelter and nesting opportunities. Considering the foraging behavior and ecological preferences of the cranes [7,15,46], it is reasonable to assume that habitats with higher NDVI values are more conducive to supporting their populations. In addition, the dense vegetation provided by higher NDVI values also contributes to the structural complexity of the habitat. This complexity is essential for red-crowned cranes as it provides optimal nesting sites and protection from predators. These characteristics are essential for successful breeding [47] and population persistence of the cranes. By selecting habitats characterized by higher NDVI values, red-crowned cranes can establish stable territories for courtship displays and breeding activities with reduced disturbance, increasing the chances of breeding success.

Annual precipitation plays a pivotal role in shaping the availability of water resources, which is critical to the survival and persistence of the red-crowned crane. These cranes exhibit a strong ecological connection to wetland habitats, particularly marshes [9]. Therefore, the annual precipitation pattern in a specific area has a significant impact on the

formation, stability, and suitability of the wetland habitats that red-crowned cranes rely on. Suitable annual precipitation can greatly contribute to the establishment and maintenance of optimal wetland habitats, thus making them more suitable for the crane's population.

Soil erosion is a dynamic process characterized by the separation, transport, and deposition of soil particles, which can adversely affect habitat. As erosion proceeds, it gradually decreases vegetation cover, resulting in a decrease in foraging opportunities and nesting suitability for the cranes. Additionally, soil erosion has the potential to disrupt the stability of nesting sites, rendering them more vulnerable to environmental factors and increasing the susceptibility of the crane population. To address these challenges, it is crucial to implement effective erosion control measures and conservation strategies. Implementing practices such as contour farming, terracing, reforestation, and the establishment of buffer zones can play a pivotal role in mitigating erosion and promoting vegetation regeneration.

The relationship between natural population growth rates and red-crowned crane habitats is complex and multifaceted. As populations grow, the demand for resources increases, which can lead to habitat loss, fragmentation, and degradation. Additionally, human activities, such as urbanization, agriculture, and infrastructure development, often result in the conversion of natural habitats into built environments or altered landscapes. These changes can directly encroach onto the crane's habitat and reduce its availability and suitability. In addition, population growth can indirectly affect the habitat through various socioeconomic [11] and cultural factors. Growing populations can drive industrial expansion and intensify resource extraction, potentially leading to pollution, deforestation, and habitat destruction. In turn, increasing human demand for food, water, and energy can lead to increased pressure on natural resources, further affecting the quality and availability of the habitat for the crane. Notably, the level of awareness, environmental education, and conservation efforts in growing human communities can play an important role in mitigating negative impacts on the crane habitat.

The Engel coefficient quantifies the proportion of total household Income allocated to expenditures on food. A higher Engel coefficient signifies lower relative food expenditures, indicating that rural residents are more economically prosperous and may have better access to resources. The positive correlation between Engel's coefficient and the crane's habitat suggests a potential link between economic prosperity and habitat quality. The higher income levels of rural residents may provide the means to invest in land management practices, conservation efforts, and the implementation of sustainable land management strategies. With increased financial resources, local communities are more likely to engage in environmental conservation actions, promote sustainable agricultural practices, and support habitat restoration projects, all of which contribute to the improvement of the crane's habitat.

Library collections can serve as indicators of environmental awareness, education, and community involvement in conservation efforts. A community with a rich library collection may exhibit a greater likelihood of prioritizing habitat conservation and implementing sustainable development, which may indirectly influence the habitat of the red-crowned crane. The use of library collections as a key indicator of influence on the crane habitat highlights the potential significance of environmental education, community engagement, and access to information in shaping conservation attitudes and behaviors.

### 4.3. Evolutionary Trends of the Population of Red-Crowned Cranes

In this study, the performance of the SVR model was found to be suboptimal. To improve the prediction accuracy, we attempted to optimize the regularization and width parameters of the SVR model using various optimization algorithms. Additionally, based on existing studies, we explored the effectiveness of manually assigning weights to the SVR model. By considering the actual distribution of the dataset, this manual weighting approach enhanced the model's generalization performance and solved the problem of unbalanced feature weights [48,49]. Moreover, the results obtained from training the model revealed fluctuations and noticeable upward movements in the population size of

cranes in Yancheng area from 2016 to 2020. The maximum population size reached 952 and an overall growth rate of 69.70%. These findings indicate that the transformation of industries and economic development in Yancheng area, along with the shift toward eco-friendly development in agriculture, industry, and housing construction, have positively influenced the habitat and living conditions of the red-crowned cranes. However, it is important to note that the volatility of the upward trend implies the potential for interannual variations in cranes' population size. Such fluctuations can be attributed to multiple factors, including environmental conditions, resource availability, breeding success, and anthropogenic activities. Therefore, it is imperative that a comprehensive analysis of potential drivers be conducted and population dynamics be continuously monitored to determine the long-term sustainability of this upward trend and to provide guidance for future conservation efforts targeting cranes in the Yancheng area.

**5. Conclusions**

This study focuses on the red-crowned crane habitat in Yancheng and uses the "EWM-AHP" comprehensive evaluation model to establish an index system for evaluating the quality of crane habitat based on the EES system in Yancheng; it also conducts a comprehensive analysis of its key influencing factors. The main conclusions are as follows:

(1) By assigning weights to the indicators of the EES system, it can be observed that the social system plays a dominant role in influencing the habitat of the cranes, among which the local population density and the public library collection have a higher weight. Additionally, the population size of red-crowned cranes in Yancheng exhibits similar fluctuations, with the largest population size in 1999 (1128 individuals) and the smallest population size in 2009 (502 individuals). The correlation between the degree of coordinated development of the EES system and the population size of the cranes was further investigated, and the Spearman correlation coefficient was 0.69. This indicates that there is an interaction between the two, and the degree of coordinated development of the EES system has a positive effect on the quality of the habitat of the cranes. However, it is worth noting that although the coordinated development degree of the EES system in Yancheng, Jiangsu Province showed an overall increasing trend with a growth rate of 61.35%, the highest development degree of the EES system in the sample period was only 0.4175, and there is still much room for improvement.

(2) The results of LASSO regression and negative binomial regression based on this fit show that three types of ecological indicators (normalized vegetation index, annual rainfall, soil erosion area) and three types of social indicators (natural population growth rate, Engel's coefficient of rural residents, public library book collection) are the key factors affecting the habitat quality of Yancheng red-crowned cranes, among which the positive effect of vegetation cover is most obvious Among them, the vegetation cover has the most obvious positive influence on the quality of the habitat, with an influence coefficient of 0.528; the natural population growth rate has the most obvious negative influence on it, with an influence coefficient of -0.034.

(3) Based on the above experiments, this study predicted the evolution trend of the population size of red-crowned cranes in Yancheng area from 2016 to 2020 by further fitting the RS-SVR model, and the results showed that the population size of the red-crowned cranes in Yancheng area showed fluctuating upward movements with a maximum population size of 952 and an overall growth rate of 69.70% during the prediction period.

**Author Contributions:** Conceptualization, Y.W., L.F., C.L., L.W. and H.X.; methodology, Y.W.; software, Y.W.; validation, Y.W. and L.F.; formal analysis, Y.W.; investigation, Y.W. and L.F.; data curation, Y.W.; writing—original draft preparation, Y.W.; writing—review and editing, Y.W. and L.F.; visualization, Y.W.; supervision, L.F.; project administration, L.F.; funding acquisition, L.F. All authors have read and agreed to the published version of the manuscript.

**Funding:** This research was funded by National Social Science Fund of China, grant number 21BGL164.

**Data Availability Statement:** All the data are from open data sources, which have been indicated in the text.

**Conflicts of Interest:** The authors declare no conflict of interest.

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
