# Peer review of "The Influential Factors of the Habitat Quality of the Red-crowned Crane: A Case Study of Yancheng, Jiangsu Province, China"

_land, doi:10.3390/land12061219_

Round 1

Reviewer 1 Report

The study on "The Influential Factors of Habitat Quality of the Red-crowned Crane: A Case Study from Yancheng, Jiangsu Province, China" is interesting if they improve their presentation. Many parts of result and conclusion should be discussion. The conclusion is too long. the Methodology was too details and had lot of equations that may not necessary.

I also attach my comment in the pdf.  

-

Author Response

Dear Professor,

Thank you very much for your comments and professional advice. These opinions help to improve the academic rigor of our article. As for the questions you raised, we have made the following modifications:

  1. The unnecessary amount you indicated in the introduction has been removed, and the structure of the remaining content has been appropriately adjusted to enhance its readability (Tables No. 1-2 below).
  2. The study area is detailed and supported by citations to the necessary literature. The name of Figure 1 is also described in more detail for understanding (No. 3-4).
  3. In response to your suggestion on the methodology section of the study, we realized that the discussion of the entropy method in the original manuscript lacked literature support, so we cited the relevant literature in the description of the method and the formula (No. 5-7).
  4. In the results section, firstly, we have rewritten the results according to your suggestion from highest to lowest (No. 8); secondly, we have added the value of the necessary statistics to support the conclusion that there is a strong relationship between the population size of red-crowned cranes and the degree of coordinated development of the EES system (No. 10); furthermore, we have added specific percentages to the conclusion that the population size of red-crowned cranes declined significantly in 1997, 2009 and 2013 (No. 10). Finally, we apologize for the poor presentation in Line 462 of the article, which attributed the value of growth of the three industries to the " comprehensive economic growth" is indeed unscientific, so we changed it to " comprehensive industrial growth value" would be more rigorous (No.13).
  5. It is worth noting that the principal component analysis used in this research is a prevalent method in multivariate statistical analysis. The core idea is to extract the primary information of the data by finding linear combinations between variables to interpret and explain the original data structure. Considering the editor's suggestion that very well-worn equations can be replaced by relevant references, we did not elaborate on this method in the research methods section of the article. However, your suggestion is valuable to improve the academic rigor of this paper, so we cited the relevant literature in the section of the article using PCA (No. 12).
  6. We have moved the results and conclusions to where they should belong and cited relevant references to enhance the discussion (No. 9,11,14,16-23).

(Please download and check the attachment, thank you)

Reviewer 2 Report

1. Figure 1. The name of Table 2 should be in the middle. 2. The citation format of the article is wrong (? Not sure, the reference should be in the upper right corner.) 3. The chart is a little confused. Change the name of the "picture + name" form. 4. Picture 3, Picture 4, the picture does not correspond to the name and location. 5. The format of the figure and the name is wrong, and the name should be placed below the figure. 6. 14 pages of pictures are piled up, slightly confused. 7. The name of Figure 5 is half above the figure, half below the figure, and should be below the figure. 8. Figures 6, 7 and Figure 5 are separated and corresponded separately. 9. The conclusion of the article structure should be before the discussion.

May be you can enhance the language touch-up.

Author Response

Dear Professor,

Thank you very much for your comments and professional advice. These opinions help to improve the academic rigor of our article. As for the questions you raised, we have made the following modifications:

(Please download and check the attachment, thank you)

Reviewer 3 Report

I question your use of trend on page 15, line 522.  Two data points do not indicate a trend.  Change the verbiage to "an annual decrease was observed"

Line 523, you state "according to predicted data ... " Does this mean that the population size is growing in a manner predicted by the model or does the model indicate the population should be growing at that predicted rate?

Check your sentence structure as several sentences, while well written, are a little confusing due to the structure.

You should change the use of since to because as since relates to a change in time.

Author Response

Dear Professor,

Thank you for your appreciation. These opinions help to improve academic rigor of our article. Based on your suggestions and request, we have made the following modifications:

Comment 1: I question your use of trend on page 15, line 522.  Two data points do not indicate a trend.  Change the verbiage to "an annual decrease was observed".

Response: Thank you for your rigorous comment. You are correct that the line between multiple data points does not strictly prove a trend. Considering that the population size of the crane population did not monotonically decline during 1990-2015 but fluctuated within a specific range, we have modified the original expression by changing "showing a fluctuating downward trend during the sample period." to "showing a fluctuating downward phenomenon with an overall decline rate of about 36.03% during the sample period."

Comment 2: Line 523, you state "according to predicted data ... " Does this mean that the population size is growing in a manner predicted by the model or does the model indicate the population should be growing at that predicted rate?

Response: In our experimental data, the period of the population size of red-crowned cranes is 1990-2015, and the time of the remaining indicators is 1990-2020. That is, these data, for example, natural population growth rate and population size during the predicting period (i.e., 2015-2020), are observed values that have been collected, because of which, in this research, we use this data as explanatory variables. The RS-SVR model aims to predict the future evolution of the population size of the red-crowned crane by using the above indicators. To sum up, population size is the objective data collected before, whose values and growth rates are consistent with the objective facts and are not influenced by this model.

Comment 3: Check your sentence structure as several sentences. while well written, are a little confusing due to the structure. You should change the use of since to because as since relates to a change in time.

Response: We apologize for the poor language of our manuscript. We have now worked on both language and readability. We hope that the flow and language level have been substantially improved.

(Please download and check the attachment, thank you)

Reviewer 4 Report

I disagree with the contents of Table 1. The authors should clarify why they chose only three factors for the ecological factors. The ecological factors should be those determining the habitat suitability of the red-crowned cranes. Using these three factors is scientifically not justified for me.

Author Response

Dear professor,

Thank you for your appreciation. As for the questions you raised, we have made the following modifications:

Comment 1: I disagree with the contents of Table 1. The authors should clarify why they chose only three factors for the ecological factors. The ecological factors should be those determining the habitat suitability of the red-crowned cranes. Using these three factors is scientifically not justified for me.

Response: We appreciate the reviewer's insightful suggestion and agree with the idea that habitat factors are essential indicators affecting the quality of habitat for the crane. However, the purpose of this study is to consider the multifaceted impacts of ecological, economic, and social systems on habitat in the Yancheng area in a holistic manner by placing the habitat quality evaluation of the red-crowned crane in the EES system in order to establish a link between the impacts of human activities on habitat quality. Therefore, this paper's starting point differs from the previous use of habitat factors to evaluate the suitability of the habitats or protected areas of the cranes. Instead, the normalized vegetation index, annual rainfall, and soil erosion area are selected to build an indicator system for the ecosystem of the Yancheng area in conjunction with the relevant studies of Peng Huang and Jingjing Dang (please refer to the fourth paragraph of the Introduction). However, your suggestion is still constructive, and we will explore it more deeply in the future study.

Thanks again for your helpful suggestions on our manuscript.

Round 2

Reviewer 1 Report

-